# Group Therapy with Peer Support Provider Participation in an Acute Psychiatric Ward: 1-Year Analysis

**DOI:** 10.3390/healthcare11202772

**Published:** 2023-10-19

**Authors:** Rosaria Di Lorenzo, Jessica D’Amore, Sara Amoretti, Jessica Bonisoli, Federica Gualtieri, Ilaria Ragazzini, Sergio Rovesti, Paola Ferri

**Affiliations:** 1Mental Health Department and Drug Abuse, AUSL-Modena, 41125 Modena, Italy; 2Nursing Programme, University of Modena and Reggio Emilia, 41125 Modena, Italy; jessica1900.jda@gmail.com; 3School of Specialization in Psychiatry, University of Modena and Reggio Emilia, 41125 Modena, Italy; sara.amoretti.25@gmail.com (S.A.); jessicabonisoli.jb@gmail.com (J.B.); f.gualtieri@ausl.mo.it (F.G.); ilaria.ragazzini8@gmail.com (I.R.); 4Department of Biomedical, Metabolic and Neural Sciences, University of Modena and Reggio Emilia, 41125 Modena, Italy; sergio.rovesti@unimore.it (S.R.); paola.ferri@unimore.it (P.F.)

**Keywords:** group psychotherapy, acute psychiatric ward, Bion’s assumptions, therapeutic factors, mentalization processes

## Abstract

(1) Background: Group psychotherapy improves therapeutic process, fosters identification with others, and increases illness awareness; (2) Methods: In 40 weekly group sessions held in an acute psychiatric ward during one year, we retrospectively evaluated the inpatients’ participation and the demographic and clinical variables of the individuals hospitalized in the ward, the group type according to Bion’s assumptions, the main narrative themes expressed, and the mentalization processes by using the Mentalization-Based Therapy-Group Adherence and Quality Scale (MBT-G-AQS); (3) Results: The “working” group was the prevailing one, and the most represented narrative theme was “treatment programs”; statistically significant correlations were found between the group types according to Bion’s assumptions and the main narrative themes (Fisher’s exact, *p* = 0.007); at our multivariate linear regression, the MBT-G-AQS overall occurrence score (dependent variable) was positively correlated with the number of group participants (coef. = 14.87; *p* = 0.011) and negatively with the number of participants speaking in groups (coef. = −16.87, *p* = 0.025); (4) Conclusion: our study suggests that the group shows consistent defense mechanisms, relationships, mentalization, and narrative themes, which can also maintain a therapeutic function in an acute ward.

## 1. Introduction

In 1905, Joseph Pratt, an internist at Boston Hospital, recognizing the importance of the psychological component in somatic disorders, for the first time introduced group therapy techniques for patients suffering from pulmonary tuberculosis. Pratt immediately recognized the benefit of group therapy, which increased participants’ compliance and success with the treatments undertaken due to sharing the common illness experience [1].

The oldest model of group psychotherapy can be traced back to Jacob Levi Moreno who, in 1921, developed the group technique called Psychodrama [2], still in use, which has its roots in theatre, psychology, and sociology. Although this technique is applied in group contexts, it actually focuses on the characteristics of each individual in staging different relational roles related to potential events or difficulties. Most of the techniques used in Psychodrama, such as role reversal, soliloquy, or double-mirroring, are employed to represent the psychological conflicts [3,4].

The common antecedent of all schools of group therapy can be considered the Freudian theory of the group. Freud was the first author who considered a group as a psychological unit characterized by identification mechanisms between members, which favor attachment to a leader or common ideals or interests. The deep and unconscious processes of relations in a group, however, necessarily imply a loss of the independence and personal identity of each individual within a group [5].

In the first half of the 1900s, the English psychoanalytic school made a great contribution to the development of group therapy theory, hypothesizing the presence of group dynamics that allow it to function as a single object which is different from the sum of the individuals who participate in it. Bion was the first author to use the term “group therapy” and to promote the concept of “basic assumptions”, which represent unconscious fantasies, through which each individual relates spontaneously and unconsciously to another in a group. These basic assumptions can shape the group communication modality, based on which the following group types have been identified: “pairing”, in which the magical belief that the resolution of all problems is possible by means of the union of two members of the group; “dependency” (in which all participants depend on a leader, who can solve all problems); and “fight-flight” (in which group participants identify an enemy who has to be attacked or from whom to escape to maintain the spirit of the group). In addition, the author identifies another two types of groups: the working group, which implies the rational cooperation of participants addressed to a common aim, and the disorganized one, in which verbal communication and effective relationships are so inconsistently aggressive and conflicting that the main communication modality cannot be identified [6,7].

### 1.1. The Group Therapy in Acute Psychiatric Setting

The acute psychiatric ward represents the preferred setting for the management of psychiatric emergencies/urgencies which need hospitalization and for the most severe psychiatric conditions. It has a central role in clinical and diagnostic evaluation, pharmacological therapies, and coordination with outpatient services. The hospitalized subjects, in general, are severely ill and also have a very difficult social background [8]. Group therapy in acute psychiatric wards is an extremely useful activity but is equally neglected and little-recognized as an integral part of inpatient treatment, probably due to the very limited literature on the subject.

The pioneer of group psychotherapy implementation in acute psychiatric wards was Irvin Yalom, who described for the first time the characteristics of this setting and identified two main difficulties in group session management: the rapid turnover of participants due to brief hospitalizations, which makes the group composition highly unstable; and the heterogeneity of inpatient psychopathology which inevitably leads to having group participants in serious psychiatric conditions, not very interested in introspection but concentrated only on refusing hospitalization, alongside other more compensated patients [8,9,10]. He researched the therapeutic factors of group therapy, identifying 12 “curative factors” or “mechanisms of change that occur through an intrinsic interplay of varied guided human experiences”: altruism, cohesion, universality, interpersonal learning input and output, guidance, catharsis, identification, family re-enactment, self-understanding, instillation of hope, and existential factors. They are now widely accepted as the main mechanisms of group psychotherapy which favor psychological changes, promoting recovery [11,12].

Other factors influencing group therapy in an acute care unit include the variability of treatments and the environment in which subjects find themselves living, suggesting that the ward system can be highly complex and influential [13].

Some authors identified the purpose of hospital group therapy as strengthening the development of the Ego and relational functioning capacity [14,15]. Following that, other therapeutic purposes of group therapy were identified: individuals’ involvement in therapeutic process, development of awareness of being able to help others, involvement of talking together as cathartic mechanism, reduction of isolation, and reduction of anxiety caused by hospitalization [8,9,12,13].

In acute psychiatric wards, the group is seen by inpatients as an advantage rather than an overload since the group increases and improves their perception of reality [9]; reinforces self-efficacy, i.e., the ability to successfully complete a task such as participating in a group session until the end; and provides the possibility to help and support others, testifying to participants the benefit of group therapy [12,15,16].

In accordance with Yalom and other authors [9,17], all these objectives are pursued inside the group from a perspective of “here and now”, considering the current situation of the individuals regardless of their reasons for hospitalization, inviting participants to communicate more clearly, to approach and relate more adequately to others, to express positive feelings, to offer support, to listen, and to reveal themselves [9,17].

The group psychotherapy work in hospital settings is radically different from outpatient-setting psychotherapy. In particular, the hospital group does not have the temporal and spatial conditions to identify the main interpersonal problems nor to observe the long-term change process. The group therapist in a hospital setting has to adopt an active posture working to facilitate some small changes within the group, using the technique of “here and now”. He performs the important function of guardian in regulating the impact of potentially negative extra-group factors, such as the extreme variability of patients sometimes discharged before their participation in the group or the devaluation of the therapeutic group within the ward, and has the duty to address these issues, ensuring that they do not permeate within the group and diminish the experience [9,13,17,18]. A meta-analytic review highlighted that beneficial effects were found for inpatient group therapy, with different effectiveness related to different psychiatric disorders [19].

More recently, other authors [20] analyzed group functioning by means of mentalization processes [21], through which we give meaning implicitly and explicitly to the others and to ourselves in terms of subjective states and mental processes [2]. Mentalization-Based Group Therapy (MBT-G) identifies in the group a cultural system of norms (matrices) in which the individual characteristics of each member can be staged and in which important events can become the object of collective reflection [22]. In accordance with Karterud, the purpose of MBT-G is to improve the participants’ ability of mentalizing “in close relationships”, and for this aim, the group is focused on “interpersonal transactions”, exploring the life events of the participants [23,24].

### 1.2. Peer Support Provider

The term “peer support provider” (PSP) indicates a person who, after having suffered from a disorder and having already recovered from it, is trained to share his/her illness experience with individuals suffering from similar disorders to improve their illness awareness and therapeutic adherence [25]. The PSP makes available his/her knowledge acquired through the direct experience of pathology, improving patients’ effective management of clinical and care services, collaboration in clinical trials, and participation in care activities [25,26]. In behavioral health, “a peer” is usually used to refer to someone who shares the experience of living with a psychiatric disorder and/or addiction.

### 1.3. Objectives of the Study

To evaluate group psychotherapy sessions implemented in an acute psychiatric ward, we analyzed the group type according to Bion’s basic assumptions, the main narrative themes, the mentalization processes, the participants’ adherence, and the role of PSPs within this setting.

### 1.4. Expected Results

We took into account the main features of subjects hospitalized in acute wards: acute psychopathological conditions, the rapid turnover of participants due to brief hospitalizations, the heterogeneity of inpatient psychopathology and diagnosis, the variability of psychopharmacological treatments, as well as the adherence to treatment. Due to these characteristics, we expected narrative themes dominated by paranoid contents, suspiciousness, and oddity; a prevalence of oppositional and disorganized groups according to Bion’s classification; and approximately 50% of inpatients participating in the group therapy having a reduced mentalization process capacity.

## 2. Materials and Methods

### 2.1. Study Design, Period, and Setting

We carried out a qualitative–quantitative retrospective study focused on psychotherapeutic group sessions implemented in the Service for Psychiatric Diagnosis and Treatment (SPDC) in the general hospital “OCSAE” in Baggiovara (Modena). The period of one year (from 1 July 2021 to 14 June 2022) was taken into consideration for a total of 40 sessions.

The participants of the group sessions were represented by all subjects hospitalized in the SPDC who voluntarily agreed to participate in the group therapy, a psychiatrist, one or two residents in psychiatry, a psychiatric rehabilitation therapist, one or two nurses, and the PSP, who did not work inside the ward but attended all group sessions.

The group sessions were held inside the ward once a week, on the same day, if possible (Tuesday), and lasted 45 min. Each session was opened by the psychiatrist with the communication of the session duration and the aims of group therapy: a confrontation of the psychological and relationship issues faced before and/or during the hospitalizations suggested by participants in full freedom. In Italy, according to the 180/78 and 833/78 Laws, the acute psychiatric ward is called the Service for Psychiatric Diagnosis and Treatment (SPDC), must be located in a general hospital, and can accommodate a maximum of 15 inpatients in voluntary and involuntary treatment.

### 2.2. Quantitative Analysis

We collected variables related to participation in group therapy: number of subjects in each group session and number of people hospitalized in SPDC; number of subjects who left a group session before it ended; and number of subjects who actively participated in the group, verbally communicating their thoughts.

The following demographic and clinical variables of individuals hospitalized in the ward at the moment of each group session were collected: average age; number of women and men; number of voluntary and involuntary hospitalizations; and psychiatric diagnoses of hospitalized subjects according to ICD-9-CM.

### 2.3. Qualitative Analysis

The qualitative analysis of the verbal contents expressed during group psychotherapy was conducted at the end of each group session by the therapists who had participated in it and consisted of two parts:Analysis of the functioning mode of the therapeutic group according to Bion’s classification [6]:“the fight-flight group”, or oppositional group, occurs when group participants ally against the therapists identified as the cause of their conditions, consider hospitalization as a detention, show mistrust towards the therapeutic processes, and/or criticize the rules of ward;”the dependency group” or psychoeducational group occurs when the group participants ask for health information (clarification on psychiatric pathologies, length of hospitalization, drugs and their side effects, criteria for compulsory medical treatment, etc.) and need to be continuously reassured and supported by therapists, showing passive and immature behavior;“the pairing group” or psychological confrontation group is characterized by hoping and waiting for rescue through two parties uniting to create the perfect solution for the participants’ current conditions, which they are not able to actively modify;“the disorganized group” in which participants, often suffering from an acute psychiatric condition, are unable to find adequate verbal communication and an effective relationship with therapists and other members, showing aggressive, conflicting, and incoherent behavior;“the work group”, different from other basic assumption groups, is based on good cooperation between participants on a topic, theme, or problem to be solved without the interference of strong emotions or destructive conflicts, showing the participants’ ability to cooperate and control their emotions.Thematic analysis related to the main narrative nuclei, using an inductive approach in 5 phases: (1) becoming familiar with the topic, (2) creation of initial codes, (3) identification of the main themes, (4) qualitative review of the main themes, (5) definition and naming of final themes. The main narrative cores were identified by thematic analysis, i.e., a systematic method for identifying, organizing, and investigating themes within a data collection by using an inductive approach [27]. We conducted the thematic analysis on the collection of data recorded by the therapists at the end of each group session.
Overall, the thematic analysis was conducted through 5 stages by the therapists who participated in the group session:Step 1. Becoming familiar with the topic: we proceeded by analytically re-reading expressed themes, becoming familiar with them in order to identify the relevant ones;Step 2. Creation of initial codes: codes can be defined as a kind of label or concise summary of the expressed themes created by interpreting both semantic and latent contents. The initial codes created were the following: inside/outside relating to the internal or external environment of the department; mistrust in the healthcare system; stigma of mental illness; mental suffering; perception of the passage of time; disease awareness; doctor–patient relationship; meaningful interpersonal relationships; emotion regulation; search for daily recreational activities; paranoid ideas; hospitalization experience; somatizations and bodily concerns;Step 3. Identifying the main themes: a theme is defined as a significant central concept or idea that recurs within multiple topics. Generating main themes was an active process of re-viewing initial codes, identifying areas of similarity or overlap between them, generating subthemes, and bringing together codes that appear to have a common characteristic so that they can describe a consistent pattern within all codes;Step 4. Review of the main themes: themes were reviewed assessing their relationship to all other themes and their ability to synthesize the most relevant and important elements in relation to the research question;Step 5. Definition and naming of main narrative themes: any emerging narrative nuclei were named based on the following 5 themes: Interpersonal relationships; Healing process; Introspective experiences; Paranoid ideas; Daily life activities. In some sessions, multiple main narrative themes emerged. Therefore, we grouped multiple main narrative themes together into the three following ones: (A) Treatment programs + Interpersonal relationship; (B) Treatment programs + Introspective experience; (C) Interpersonal relationship + Introspective experience. Finally, we collected a total of 8 main narrative themes.

### 2.4. Psychometric Analysis for Mentalization

We measured the level of mentalization in the group by completing the Mentalization-Based Therapy-Group Adherence and Quality Rating Scale (MBT-G-AQS), which evaluates 19 items in terms of occurrence and quality [27]: (1) Managing group boundaries; (2) Regulating group phasing; (3) Initiating and fulfilling turn taking; (4) Engaging group members in mentalizing external events; (5) Identifying and mentalizing events in the group; (6) Caring for the group and each member; (7) Managing authority; (8) Stimulating discussions about group norms; (9) Cooperation between co-therapists; (10) Engagement, interest, and warmth; (11) Exploration, curiosity, and not-knowing stance; (12) Challenging unwarranted beliefs; (13) Regulation of emotional arousal; (14) Acknowledgment of good mentalization; (15) Handling pretend mood; (16) Handling psychic equivalence; (17) Focus on emotions; (18) Stop and rewind; (19) Focus on the relationship between therapists and patients. For items 6, 7, 10, 13, and 15, only quality assessment is required.

Occurrence (score from 0 to 30) indicates how often therapists recall the mentalization process to patients in the group session; quality (score from 1 = very poor to 7 = excellent according to a 1–7 Likert scale; 0 = not applicable) indicates the level of skill and expertise of the therapist in handling the item content during the group session. The overall rating of occurrence and quality should indicate the global functioning of the group as grounds for mentalizing [27].

### 2.5. Statistical Analysis

All data collected were entered into an electronic database, and statistical analysis was carried out using STATA 2011. The data are summarized using the following tools:For continuous variables: mean, standard deviation;For categorical variables: percentages, chi2 test, and Fisher’s exact text;Forward and backward multiple linear regression was applied between the overall occurrence score of MBT-G-AQS quality (dependent variable) and the following selected variables as independent ones: number of subjects who attended groups, number of subjects who intervened in the group session by speaking, number of subjects who abandoned the group session before the end, type of group, and main narrative cores.

A usual significance level of *p* < 0.05 was considered.

### 2.6. Ethical Considerations

The approval and authorization to conduct this study were obtained from the Ethics Committee of the Wider Emilia North Area (prot. N.: 317/2022/OSS*/AUSLMO of 07/05/2022) and AUSL of Modena (prot. AOU 0020166/22), respectively.

## 3. Results

### 3.1. Quantitative Analysis

In the period considered, from 1 July 2021 to 14 June 2022, 239 patients were hospitalized in our SPDC. The demographic and clinical characteristics of the subjects hospitalized are reported in Table 1. The age ranged between 14 and 78 years, with a mean of 39.2 ± 3.8 SD. Males and females were equally represented, with a slight majority of men (n = 126; 52.7%) over women (n = 113; 47.3%). More than half of subjects (n = 122; 51%) were voluntarily hospitalized, whereas 40.2% (n = 96) were involuntarily hospitalized, with a mean of 3.9 ± 1.4 SD at the time of each session. The most common psychiatric diagnoses, according to the ICD-9-CM classification, were represented by schizophrenia spectrum disorders (6.7 ± 8.1 SD), followed by bipolar disorders (2.4 ± 1.2 SD) and personality disorders (1.9 ± 1.6 SD). The mean duration of hospitalizations was found to be 11.2 ± 2.1 SD days.

### 3.2. Qualitative Analysis According to Bion’s Assumption Classification

The 40 sessions were divided, according to Bion’s assumption classification, into 5 types of therapeutic groups, as described in Table 2. Most of the group sessions (n = 24; 60%) were represented by “working” groups, in which the patients rationally confronted each other in a cooperative attitude. These groups were attended by 7.5 subjects on average (±2.1 SD), who represented more than half of all inpatients (13.3 m ± 1.9 SD); almost all the group participants verbally communicated in the group (6 ± 1.3 SD), whereas a small number of participants left the group before the session ended (mean 0.4 ± 0.7 SD). “Fight and flight” groups represented 13% of all sessions, with fewer participants who intervened by speaking in the group (5.2 ± 1.9 SD), and many more patients left the session before its end (1.2 ± 2. 7 DS). Another 13% of groups presented a “disorganized” modality whereas “dependency” (8%) and “pairing” (8%) groups were less frequent (Table 2).

We found a positive statistically significant correlation (Spearman’s ρ = 0.572, *p* = 0.0001) between the number of patients admitted to the ward and the number of patients participating in the group (Table 2). Similarly, the number of group participants was positively statistically significantly correlated (Spearman’s ρ = 0.571, *p* = 0.0001) to the number of participants who actively intervened in the group sessions by speaking.

### 3.3. Qualitative Analysis According to the Main Narrative Themes

From the initial 40 narrative themes, through 12 codes, we found five main narrative cores (Table 3), with the following frequency in the 40 group sessions (Table 4): treatment programs (19/40; 47.5%), interpersonal relationships (4/40; 10%), introspective experiences (4/40; 10%), paranoid ideas (3/40; 7.5%), and daily activity functioning (2/40; 5%). In some group sessions, more than one main narrative core was highlighted: treatment programs and interpersonal relationships (3/40; 7.5%), treatment programs and introspective experiences (3/40; 7.5%), and interpersonal relationships and introspective experiences (1/40; 2.5%). It was not possible to identify a main theme in only one session.

We found a statistically significant correlation between the types of groups and the main narrative cores (Fisher’s exact, *p* = 0.007): in “working” groups, the more prevalent narrative core was represented by “treatment programs” whereas in “fight and flight” groups, the most frequent theme was represented by “paranoid ideas” (Table 4).

### 3.4. Psychometric Analysis by Completing the Mentalization-Based Therapy-Group-Adherence and Quality Rating Scale (MBT-G-AQS)

The MBT-G-AQS overall score for occurrence was 15.46 ± 4.65 SD and for quality was 4.60 ± 0.9. As shown in Table 5 and Figure 1, item 3, “Initiating and fulfilling turn taking”, was the most recurrent with a quality score of 5.6 ± 1.17 (Table 5, Figure 1). The MBT-G-AQS scale reliability was confirmed by an alpha coefficient > 0.85 both in the occurrence and in the quality of items.

### 3.5. Multiple Linear Regression Model

At our forward and backward stepwise multiple linear regression between the overall occurrence score of the MBT-G-AQS and other variables (R-squared = 0.18; Adj R-squared = 0.13), we observed a positive statistically significant correlation with the number of participants in the group and a negative one with the number of participants who actively intervened in the group by speaking, as shown in Table 6.

## 4. Discussion

Our study was focused on the evaluation of a therapeutic group implemented in an acute psychiatric ward by means of different epistemological approaches: the main narrative themes, the type of group, patient participation, and a PSP role evaluation within this setting. The acute psychiatric ward where the therapeutic group was implemented was characterized by the prevalence of subjects with severe psychiatric disorders in an acute phase, as evidenced by most diagnoses being schizophrenic spectrum disorders and bipolar disorders, by the non-adherence to treatment in about 29% of involuntary admissions during the observation period, and by the high patient turnover due to the brief duration of hospitalizations (average length of hospitalization = 11.2 days), in line with data available in the literature [27,28]. As emphasized by Irvin Yalom [9], this setting, characterized by urgency, severity, a heterogeneity of psychiatric pathologies, and variability in the psychopharmacological treatments, as well as non-adherence to treatment, cannot in itself favor the implementation of group psychotherapy, in which the continuity of setting and the voluntary participation in the psychotherapeutic group are necessary. We can hypothesize that in our ward, the continuity of the group setting was guaranteed not only by pre-established time and place schedules but also by the regular presence of a PSP [29], who, maintaining the so-called group memory and acting as a glue with the outside, offered to group participants a model of empathic identification. The role of the PSP was to bring to the group his/her experience concerning mental disorder and a patient–therapist relationship to help patients reduce anxiety and helplessness towards their illness [30].

Our study allowed us to evaluate the therapeutic group through different epistemological modalities of interpreting, classifying, and defining it: Bion’s classification based on basic assumptions, the narrative themes analyzed in accordance with Yalom’s therapeutic factors, and the mentalization processes applied to group activities, according to Fonagy and Allison [22].

We classified the group typologies according to Bion’s basic assumptions into the “pairing” or psychological confrontation group, “dependency” or psycho-educational group, “fight-flight” or contrapositive group, disorganized group, and work group. In our study, the prevailing typology was the work group, suggesting a rational connotation of group activity, which can be interpreted as an attempt by professionals to overcome the regressive behavior of inpatients and, at the same time, to avoid overwhelming emotional contents to organize the patients’ experiences in a more detached manner. Our study confirms the findings of a previous study conducted in the same setting, in which, despite the ward overcrowding and the refusal of care expressed by patients, the “work” group modality was prevalent [31]. Due to the features of an acute hospital context, we would expect a large number of sessions to be dominated by Bion’s basic assumptions, real defenses, and unconscious resistances adopted by the group towards the treatment. In contrast, the results of this study show how the prevailing typology was the ‘work’ group, suggesting the therapeutic value of group sessions even in a hospital setting.

The number of patients who interrupted the group session indirectly highlighted the scarce cohesion of the group and was consistent with the typology of the groups; a greater number of patients interrupted the “fight-flight” group sessions, characterized by paranoid ideas and conflicting attitudes, whereas a reduced number of subjects decided to leave “work” groups.

We highlighted a correlation between the type of group according to the basic assumptions and the main narrative cores: “fight-flight” groups, in which the subjects allied themselves against the therapists identified as the cause of their condition, were characterized by the prevalence of paranoid ideas; “work” groups were characterized by the discussion of treatment programs; in “dependence” groups, introspective experiences were prevalently recorded, and in “pairing” groups, interpersonal relationships were recorded, both dominated by regressive psychological attitudes. In disorganized groups, mixed themes, treated in a fragmentary manner, were recorded, and in one of these groups, no main theme was recorded, suggesting that participants were unable to find shared modes of verbal communication and effective relationships [6].

Our results suggest that basic assumptions (“fight and flight”, “dependence”, “pairing”, etc.) in group sessions hindered the development processes, similarly to the defense mechanisms that patients implement in individual sessions of psychotherapy.

The number of patients who left the group session before conclusion indirectly reflected the cohesion of the group and was consistent with the typology of the groups; in fact, a greater number of patients moved away from the “fight and flight” groups, identifying in this basic assumption the one in which adherence to the group is absolutely lacking. In self-report studies on Yalom’s therapeutic factors, “cohesion” is considered the main therapeutic factor which facilitates the other therapeutic factors [29]. More recent authors have confirmed that “cohesion” can represent one of the prevalent therapeutic factors in a group [18,32,33].

We compared the main narrative themes expressed in our groups and the eleven therapeutic factors identified by Yalom. In particular, we hypothesize that Yalom’s factors “instillation of hope” and “source of information” could overlap with our narrative core “treatment pathway”, since it includes both future planning and clarifications about hospitalization and psychopathological symptoms. In the sessions in which the theme of interpersonal relationships emerged, many of Yalom’s therapeutic factors could be represented: altruism, corrective revision of the primary family group, development of socialization techniques, imitative behavior, and interpersonal learning. Our narrative core ‘introspective experiences’ could include dynamics related to ‘catharsis’, described by Yalom as the achievement of a greater awareness of one’s own mental illness. The paranoid ideas we detected do not fall within the group therapeutic factors identified by Yalom, representing instead acute psychopathological symptom dimensions based on dysfunctional dynamics [9].

The overall occurrence score of the MBT-G-AQS suggests that therapists sufficiently stimulated group participants to explore their mental state and that group functioning was well-regulated, particularly the turn-taking, fostering mentalization processes. Similarly, the overall quality score of the MBT-G-AQS suggests an adequate handling of item contents by the group therapists. We showed a positive correlation at the multiple linear regression between the total recurrence score of the MBT-G-AQS items and the number of participating patients, suggesting, in accordance with the construct of the scale itself, that increased numbers of group members increased the need of recurrence group rules concerning boundaries, norms, turn taking, etc. in the therapeutic context. We also found a negative correlation between item recurrence and the number of patients who actively participated in groups, indicating that the more subjects are actively involved in group therapy, the less need there is to repeat group rules as the group is more cohesive.

### Limitations and Advantages of the Study

Among the limitations, we can enumerate the small sample (40 groups), the lack of a full transcription or recording of group sessions, the subjectivity of qualitative data (type of group, main narrative cores) which reduces the validity of causal inference, and the monocentric study which limits its generalization. Furthermore, the lack of group efficacy indicators in hospital treatments could represent another limitation, which other studies could fill. Finally, one of the limitations of the study is the lack of a control group without the peer support which could have highlighted the role of this figure in preserving the continuity of the group setting, fostering the stability of group psychotherapy.

Among the advantages, we can put in evidence the evaluation of group therapy in an acute hospital setting; the evaluation of the role of PSPs in acute ward groups; and the comparison of different ways of defining, interpreting, and classifying therapeutic group work, analyzing narrative themes and mentalization mechanisms.

## 5. Conclusions

Our study analyzed the dynamics of groups using different epistemological lenses of interpretations, classifications, and definitions, which permitted us to extrapolate coherent correlations between them, suggesting that a group can represent a *unicum* organism in terms of defense mechanisms, relationships, mentalization processes, and narrative themes expressed, which are consistent with each other.

Due to the features of the acute hospital context, we would expect a larger number of sessions dominated by Bion’s basic assumptions, real defenses, and unconscious resistances adopted by the group towards treatments. In contrast, the results of this study show how within our SPDC, the prevailing group typology was that of “work”, suggesting the therapeutic value of group sessions even in a hospital setting. We presume that the involvement of a PSP might have contributed to the regular group activity also in a setting shaped by urgencies, acting as an intermediary with the outside world and, at the same time, maintaining the so-called memory of group sessions.

Our study confirms that group therapy in acute psychiatric wards, often neglected and little recognized as an integral part of care, can be an extremely useful activity for integrating individual psychopharmacological and psychotherapeutic treatments in order to promote processes of mentalization of events inside and outside the group and, at the same time, to develop subjects’ relational and adaptive skills.

## Figures and Tables

**Figure 1 healthcare-11-02772-f001:**
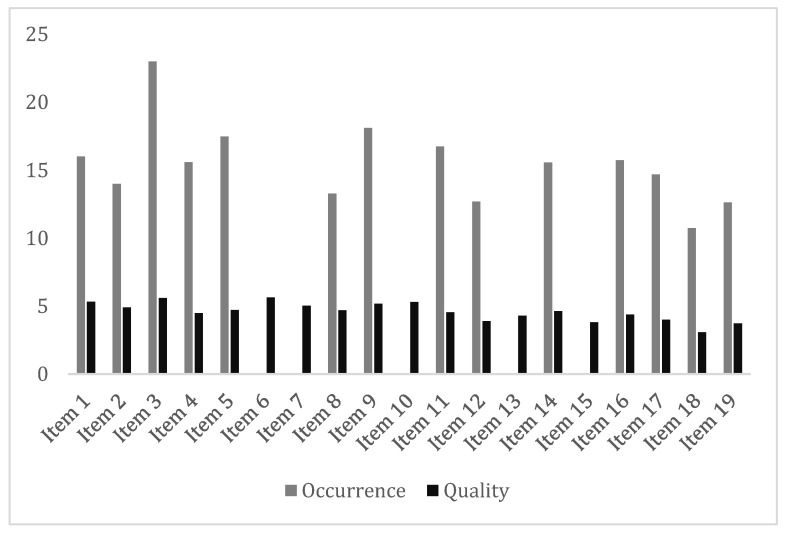
MBT-G-AQS mean scores of item occurrence and quality.

**Table 1 healthcare-11-02772-t001:** Demographic and clinical variables of subjects hospitalized at the time of the 40 group sessions.

Variables	M ± SD
Age (years)	39.2 ± 3.8
N° women	6.0 ± 2.0
N° men	7.3 ± 2.0
N° subjects involuntary hospitalized	3.9 ± 1.4
Schizophrenia spectrum disorders	6.7 ± 1.8
Bipolar disorders	2.4 ± 1.2
Personality disorders	1.9 ± 1.6
Intellective disability	0.6 ± 0.6
Organic psychotic conditions	0.2 ± 0.5
Dysthymia and depressive disorders	0.4 ± 0.6
Anorexia	0.3 ± 0.6
Others	0.8 ± 0.8
Duration of hospitalization (days)	11.2 ± 2.1

**Table 2 healthcare-11-02772-t002:** Groups subdivided according to Bion’s classification.

Type of Group n (%)	Subjects Hospitalized at the Time of Group Sessions (M ± SD)	Subjects Who Attended Group Sessions (M ± SD)	Subjects Who Intervened in the Group Sessions by Speaking (M ± SD)	Subjects Who Abandoned the Group Session before the End (M ± SD)
Fight and flight 5 (13%)	13.8 ± 3.1	8.2 ± 2.7	5.2 ± 1.9	1.2 ± 2.7
Dependency 3 (8%)	13.7 ± 0.5	7.7 ± 0.5	5 ± 1.6	0
Pairing 3 (8%)	14.3 ± 1.2	6.7 ± 2.9	4.3 ± 2.3	0.7 ± 0.6
Disorganized 5 (13%)	12.2 ± 2.2	7.6 ± 2.3	5 ± 2.3	0
Working 24 (60%)	13.3 ± 1.7	7.2 ± 2.1	6 ± 1.3	0.4 ± 0.7
Total 40 (100%)	13.4 ± 1.9	7.4 ± 2.1	5.6 ± 1.6	0.4 ± 1.0

**Table 3 healthcare-11-02772-t003:** Qualitative thematic analysis of the groups.

N° Group	Initial Narrative Themes	Codes	Main Narrative Themes
1	Inside/outside the ward Ambivalence toward safe places	Inside/outside	Healing process
2	Projection into the future: fear of discharge Projection into the past: not processing the past	Inside/outside Time	Healing process Introspective experiences
3	Communication Somatization	Healing process Psychic suffering	Healing process Introspective experiences
4	Anger both as what led to hospitalization and as experienced during the stay in the ward and which can characterize coexistence with others	Emotion regulation	Introspective experiences
5	“Feeling that I have suffered an injustice”, “we acted as guinea pigs”, “I was tied up”, “little trust in the doctor”	Distrust in the healing system	Paranoid ideas
6	Fear and psychic suffering “How others see us, how others feel our suffering that we think only we have”	Stigma Psychic suffering	Interpersonal relationships Introspective experiences
7	Time lived/time perceived	Time	Introspective experiences
8	Why and how the crisis comes: “it’s so fast that we can’t understand”	Disease awareness	Healing process
9	Difficulty in accepting the diagnosis	Disease awareness	Healing process
10	Anxiety about the outside world and a need to research a safe place	Inside/outside	Healing process
11	Life “outside the ward”	Inside/outside	Healing process
12	Stigma related to mental illness	Stigma	Interpersonal relationships
13	Boundary and vulnerability, strong mirroring among all group participants	Inside/outside Disease awareness	Healing process
14	Illness onset also understood as something yet to be “disposed of”, mentalization	Disease awareness	Healing process
15	Report of help from the group, need for support from others	Interpersonal relationships	Interpersonal relationships
16	Hospitalization	Healing process	Healing process
17	It was hard to keep a main theme	/	No main theme
18	Violence	Violence	Interpersonal relationships
19	Hospitalization	Healing process	Healing process
20	Understanding the reasons for hospitalization Attributing meaning to hospitalization through relationships with others	Disease awareness Interpersonal relationships	Healing process Interpersonal relationships
21	Length and utility of hospitalization	Healing process	Healing process
22	Healing process	Healing process	Healing process
23	Hospitalization experience	Healing process	Healing process
24	Wishes and expectations with respect to the Christmas holidays. Correct lifestyles (nutrition and physical activity)	Psychoeducation	Daily life activities
25	Internal and external persecutors	Paranoid ideas	Paranoid ideas
26	“Theory of the scapegoat”: patients—do they reinforce each other in a paranoid sense. Only one patient criticizes her aggressive behavior saying she lost control (“because provoked”)	Paranoid ideas	Paranoid ideas
27	Difficulty in finding a job (“the job that can’t be found”)	Stigma	Interpersonal relationships
28	Relationship with the outside The judgment of others Disease awareness	Inside/outside Stigma Disease awareness	Healing process Interpersonal relationships
29	Resignification of the “hospitalization” moment	Healing process	Healing process
30	Subjectivity of psychiatric disorder, patient’s point of view and doctor’s point of view	Disease awareness	Healing process
31	Moods	Emotion regulation	Introspective experiences
32	Ward criticalities	Healing process	Healing process
33	Physical activity and its benefits	Psychoeducation	Daily life
34	Hospitalization utility	Healing process	Healing process
35	Managing your own mental health	Disease awareness	Healing process
36	Importance of a good psychiatric team	Healing process/patient–doctor relationship	Healing process
37	Cause of discomfort leading to hospitalization Trust in the possibility of being able to recover and that there may be a way out	Disease awareness	Healing process
38	Isolation and trust	Stigma Path of care/ doctor–patient relationship	Healing process Interpersonal relationships
39	Life experiences with considerable suffering	Psychic suffering	Introspective experiences
40	Tension and tension management strategies	Disease awareness Psychic suffering	Healing process

**Table 4 healthcare-11-02772-t004:** Main narrative cores and types of groups according to Bion’s classification.

Main Narrative Cores *	Types of Groups According to Bion’s Classification *					Total
	Fight and Flight	Dependency	Pairing	Disorganized	Working	
Treatment programs	0	1	1	1	16	19
Paranoid ideas	2	0	0	0	1	3
Interpersonal relationships	1	0	2	0	1	4
Introspective experiences	0	1	0	1	2	4
Daily activities	0	0	0	1	1	2
(A) Treatment programs + Interpersonal relationship	0	0	0	1	2	3
(B) Treatment programs + Introspective experience	1	1	0	0	1	3
(C) Interpersonal relationship + Introspective experience	1	0	0	0	0	1
No narrative core identified	0	0	0	1	0	1
Total	5	3	3	5	24	40

* Main narrative cores vs. Types of groups according to Bion’s classification: Fisher’s exact, *p* = 0.007.

**Table 5 healthcare-11-02772-t005:** MBT-G-AQS item occurrence and quality score.

N.	Item	Occurrence (Range 0–30) M ± SD	Quality (Range 0–7) M ± SD
1	Managing group boundaries	16.03 ± 8.19	5.33 ± 1.38
2	Regulating group phasing	14 ± 7.89	4.9 ± 1.41
3	Initiating and fulfilling turn taking	23 ± 6.58	5.6 ± 1.17
4	Engaging group members in mentalizing external events	15.6 ± 7.93	4.48 ± 1.80
5	Identifying and mentalizing events in the group	17.5 ± 7.28	4.73 ± 1.40
6	Caring for the group and each member	/	5.65 ± 1.37
7	Managing authority	/	5.03 ± 1.95
8	Stimulating discussions about group norms	13.3 ± 8.25	4.69 ± 1.79
9	Cooperation between co-therapists	18.13 ± 7.90	5.18 ± 1.48
10	Engagement, interest, and warmth	/	5.3 ± 1.54
11	Exploration, curiosity, and not-knowing stance	16.75 ± 8.29	4.55 ± 1.78
12	Challenging unwarranted beliefs	12.7 ± 7.62	3.9 ± 1.93
13	Regulation of emotional arousal	/	4.3 ± 1.77
14	Acknowledgment of good mentalization	15.58 ± 7.31	4.63 ± 1.58
15	Handling pretend mood	/	3.82 ± 2.02
16	Handling psychic equivalence	15.75 ± 6.76	4.38 ± 1.53
17	Focus on emotions	14.7 ± 8.55	4 ± 1.89
18	Stop and rewind	10.75 ± 8.59	3.08 ± 2.05
19	Focus on the relationship between therapists and patients	12.63 ± 8.84	3.74 ± 1.86
Total		15.46 ± 4.65	4.60 ± 0.91

**Table 6 healthcare-11-02772-t006:** Multiple linear regression (forward and backward stepwise model) between the overall occurrence score of the MBT-G-AQS and the other variables.

Variables	Coef.	Conf. Int. 95%	Probability
MBT-G-AQS Overall Occurrence Score			
Subjects who attended groups	14.87	3.57; 26.18	*p* = 0.011
Subjects who intervened in the group session by speaking	−16.87	−31.52; 2.23	*p* = 0.025

## Data Availability

Not applicable.

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
