# Peer review of "Group Therapy with Peer Support Provider Participation in an Acute Psychiatric Ward: 1-Year Analysis"

_healthcare, 2023, doi:10.3390/healthcare11202772_

Round 1

Reviewer 1 Report

First of all, I’d like to thank the authors for their work and efforts in tackling such a topic of improving the health of mentally ill patients. This publication will help to improve the rehabilitation processes and is important.

Below, please, consider some points to improve your manuscript.

General:

1)      A title could be improved since it is not a Methods article. It is better to appoint to the results of what you explore rather than in methods.

2)      Check for extra/missing spaces to remove and missing punctuation – For example: a final stop is missing line 107, p.3.

3)      And grammar and words use: should be plural “others” instead of “other” in Introduction (lines 130-132, p.3):

More recently, other authors [20] analyzed the group functioning by means of mentalization processes [21], through which we give meaning implicitly and explicitly to the other and to ourselves in terms of subjective states and mental processes [2] (132-135).”

What does the numbers (132-135) mean)?

Abstract and further (if relevant):

1)      Please, put official all statistical abbreviations and report exact “coef” if it was Pearson, r or Spearman, rho, etc…

Results:

2)      Table 2: No legend provided to signs used as upper symbols in the last row

3)      Table 2: The last row: you have a number in the “hospitalised” 2nd column total less than in line 3 – Dependency. Is it correct? In the first column % you provide, what they stand for? (not very clear)

4)      What do you mean by “lite” in the concept: “Lite “outside the ward” (line 11, Table 3) and “POV” in line 30, Table 3.

5)      Table 3, line 35: a typo: “health” should be instead of “healt”

6)      “p-value is missing in the report of results: “ We found a statistically significant correlation between the types of group and the 254 main narrative cores (Fisher's exact = 0.007): …” (line255, p.9)

7)      Fig. 1 is duplicating the results represented in the Table 5. How “Quality” was calculated? (Should be described in Methods)

t should be checked

Author Response

Reviewer 1

First of all, I’d like to thank the authors for their work and efforts in tackling such a topic of improving the health of mentally ill patients. This publication will help to improve the rehabilitation processes and is important.

Below, please, consider some points to improve your manuscript.

Response to Reviewer 1

Thank you for your positive comments.

General:

1)      A title could be improved since it is not a Methods article. It is better to appoint to the results of what you explore rather than in methods.

1)    Response to Reviewer 1. Thank you for your meaningful observation. The title has been changed as follows: “Group therapy with peer support provider participation in an acute psychiatric ward: 1-year analysis”.

2)      Check for extra/missing spaces to remove and missing punctuation – For example: a final stop is missing line 107, p.3.

2)     Response to Reviewer 1. Done. Thank you for your suggestions.

3)      And grammar and words use: should be plural “others” instead of “other” in Introduction (lines 130-132, p.3):

More recently, other authors [20] analyzed the group functioning by means of mentalization processes [21], through which we give meaning implicitly and explicitly to the other and to ourselves in terms of subjective states and mental processes [2] (132-135).”

What does the numbers (132-135) mean)?

3)  Response to Reviewer 1. Thank you for your precise remarks. We used the plural and deleted “(132-135)”, which was a mistake.

Abstract and further (if relevant):

1)      Please, put official all statistical abbreviations and report exact “coef” if it was Pearson, r or Spearman, rho, etc

 1) Thank you for your observation. We explained that “coef”  referred to multivariate linear regressions.

Results:

2)      Table 2: No legend provided to signs used as upper symbols in the last row.

2)    Response to Reviewer 1. Thank you for your precise remark. We deleted the upper symbols in the last row.

3)      Table 2: The last row: you have a number in the “hospitalised” 2nd column total less than in line 3 – Dependency. Is it correct? In the first column % you provide, what they stand for? (not very clear).

3)    Response to Reviewer 1. We reported the mean of hospitalized subjects. Therefore, the reported data are different in the different type groups. In the last row of the table, the analysis of total groups is reported.

4)      What do you mean by “lite” in the concept: “Lite “outside the ward” (line 11, Table 3) and “POV” in line 30, Table 3.

4)    Response to Reviewer 1. Thank you for your suggestions. “Lite” is mistake and has been changed to “Life”. POV means Point of View. We changed the acronym to the full terms: “patient's point of view and doctor's point of view”.

5)      Table 3, line 35: a typo: “health” should be instead of “healt”.

5)    Response to Reviewer 1. Thank you for your suggestions. “healt” is mistake and has been changed to “health”.

6)      “p-value is missing in the report of results: “ We found a statistically significant correlation between the types of group and the 254 main narrative cores (Fisher's exact = 0.007): …” (line255, p.9).

6)     “p-value” has been reported.

7)     Fig. 1 is duplicating the results represented in the Table 5. How “Quality” was calculated? (Should be described in Methods).

7)   Thank you for your observation.  I added the following explanation of the MBT-G-AQS in the Methods:

       “We measured the level of mentalization in the groups by completing the Mentalization-Based Therapy-Group-Adherence and Quality Rating Scale (MBT-G-AQS), which  evaluates 19 items in terms of  occurrence and quality  [27]: 1) Managing group boundaries, 2) Regulating group phasing, 3) Initiating and fulfilling turn taking, 4) Engaging group members in mentalizing external events, 5) Identifying and mentalizing events in the group, 6) Caring for the group and each member, 7) Managing authority, 8) Stimulating discussions about group norms, 9) Cooperation between co-therapists, 10) Engagement, interest and warmth, 11) Exploration, curiosity and not-knowing stance, 12) Challenging unwarranted beliefs, 13) Regulation of emotional arousal, 14) Acknowledging of good mentalization, 15) Handling pretend mood, 16) Handling psychic equivalence, 17) Focus on emotions, 18) Stop and rewind, 19) Focus on the relationship between therapists and patients. For items n. 6, 7, 10, 13 and 15 only quality assessment is required.

       Occurrence (score from 0 to 30) indicates how often the therapists recall the mentalization process to patients in the group session; quality (score from 1=very poor to 7=excellent according to a 1-7 Likert scale; 0=not applicable) indicates the level of skill and expertise of therapist in handling the item content during the group session. The overall rating of occurrence and quality should indicate the global functioning of the group as ground for mentalizing [27].”

       Thank you for having taken so much time to carefully review our manuscript. I hope that the changes we have made are sufficient to improve the manuscript and make it acceptable for publication.

Best regards

Rosaria Di Lorenzo

Reviewer 2 Report

In their manuscript Mixed method analyses of 1-year group therapy with peer support provider participation in an acute psychiatric ward, the authors report on the characteristics of a psychotherapy group in an acute psychiatric ward from different qualitative and quantitative  viewpoints. The work gives insight into group therapy processes of action, making it interesting for practitioners. I generally appreciate the authors’ endeavor and find that the manuscript warrants for publication. However, I found it difficult to read and understand and would like to suggest addressing the following points before publication:

1.       The Abstract is very difficult to understand. It entails several constructs and terms and I had to read through large sections of the Introduction to understand what was being done in the study. This can be improved (provide brief definitions of constructs, remove sentences not absolutely needed, describe methods and conclusions more precisely), but it is also, in my opinion, a symptom of a bigger point:

2.       The article is not very focused. Instead, the authors use different concepts to investigate the group therapy and do not seem to have enough room to explain the rationale and methods for each of these concepts. For example, the authors state that Bion’s functioning mode of each group was assessed by a professional attending the group, but there is no description how this analysis was conducted.

3.       The quantitative analysis seems very opaque to me; from the descriptions provided by the authors, it is not possible to gather any information about the questionnaire, how many items it entailed, what constructs were assessed; about the statistical analyses, not enough details are provided to reproduce the procedures. This also makes it impossible for me to understand Figure 1. Further, it seems like a larger number of regression models was conducted, but only the significant results are reported. To me, this seems to lack adherence to general principals of good scientific practice in quantitative research methods. By conducting multiple analyses, the alpha error is increased and the p-value should be corrected by this; if the analysis is intended as an exploratory analysis, this may not be needed, but analyses should be marked as such and all results, irrespective of significance, should be reported.

4.       How were the narrative themes investigated, who decided for these and by which criteria?

5.       In the discussion, the authors communicate a sense of positive surprise; if I read this right, given the high fluctuation and strong variance in diagnoses in the psychotherapy group within the acute psychiatric ward, it might be expected that groups would show little coherence and distrust towards the practitioner might be predominant, but on the contrary, the results demonstrate a fruitful working environment. This is a finding inspiring trust in therapy groups in this setting and I appreciate it. However, mentioning these expectations only in the discussion seems to be a waste, I expect that the manuscript could benefit if the authors communicated their anticipations already in the Introduction. Typically, one would expect a hypotheses-section containing such expectations.

6.       The authors report that a peer support person helped the group maintain a positive work environment. While I support this notion and could imagine very well that this is a helpful factor, no comparison group without the peer is available. Thus there is no way of telling how much of the group’s characteristics were fostered by the presence of the peer. If no comparison group is available, this cannot be helped, but should be discussed.

In summary, it appears to me that this manuscript might benefit greatly from trimming it and removing concepts and analysis parts that are not essential in favor of the more important parts, which could then be described much more appropriately.

To my mind, the use of scientific English was adequate; minor spell and punctuation checks will be sufficient. 

I have noticed lacking commas in enumerations throughout the manuscript ("a, b, and c"; or "a, b, or c").

Author Response

Reviewer 2

In their manuscript Mixed method analyses of 1-year group therapy with peer support provider participation in an acute psychiatric ward, the authors report on the characteristics of a psychotherapy group in an acute psychiatric ward from different qualitative and quantitative viewpoints. The work gives insight into group therapy processes of action, making it interesting for practitioners. I generally appreciate the authors’ endeavor and find that the manuscript warrants for publication. However, I found it difficult to read and understand and would like to suggest addressing the following points before publication:

1) The Abstract is very difficult to understand. It entails several constructs and terms and I had to read through large sections of the Introduction to understand what was being done in the study. This can be improved (provide brief definitions of constructs, remove sentences not absolutely needed, describe methods and conclusions more precisely), but it is also, in my opinion, a symptom of a bigger point:

Response to Reviewer 2: Thank you for your meaningful suggestion. We changed the Abstract in order to clarify it as you suggested:

    “Background: Group psychotherapy improves therapeutic process, fosters identification with others and increases illness awareness; (2) Methods: we retrospectively collected 40 weekly group sessions in one year held in an acute psychiatric ward with inpatients who agreed to participate, three ward staff professionals and one expert outpatient; we evaluated the inpatients’ participation and demographic and clinical variables of individuals hospitalized in the ward;  we analyzed the group type according to Bion’s assumptions and the main narrative themes expressed; we assessed the mentalization processes by using Mentalization-Based Therapy-Group Adherence and Quality Scale (MBT-G-AQS); (3) Results: the “working” group was the prevailing one and the most represented narrative theme was “treatment programs”; statistically significant correlations were found between group types according to Bion’s assumptions and the main narrative themes (p=0.007); at our multivariate linear regression, the MBT-G-AQS overall occurrence score (dependent variable) was positively correlated with the number of group participants (coef.=12.04; p=0.047) and negatively with participants speaking in groups although in not statistically significant way (coef.=-11.12); (4) Conclusions: our study suggests that group is an unicum in terms of defense mechanisms, relationships, mentalization and narrative themes, which can maintain a therapeutic function also in an acute ward.”

2) The article is not very focused. Instead, the authors use different concepts to investigate the group therapy and do not seem to have enough room to explain the rationale and methods for each of these concepts. For example, the authors state that Bion’s functioning mode of each group was assessed by a professional attending the group, but there is no description how this analysis was conducted.

Response to Reviewer 2: Thank you for your thoughtful suggestion. We added the following description in the Method section regarding our analysis as you suggested:

    ” 1.Analysis of the functioning mode of the therapeutic group according to Bion's classification [6]: oppositional group (based on the basic assumption “The fight-flight group”) occurs when group participants ally against the therapists identified as the cause of their conditions, consider hospitalization as a detention, show mistrust to-wards the therapeutic processes and/or criticize the rules of ward; psychoeducational group (“The dependency group”) occurs when the group participants ask for health information (clarification on psychiatric pathologies, length of hospitalization, drugs and their side effects, criteria for compulsory medical treatment, etc.) and need to be continuously reassured and supported by therapists, showing passive and immature behaviour; psychological confrontation group (“The pairing group”) is characterized by hoping and magical waiting for rescue through two parties uniting to create the ‘perfect’ solution for participants’ current conditions, which they are not able to actively modify; the disorganized group in which participants, often suffering from an acute psychiatric condition, are unable to find  adequate verbal communication and an effective relationship with therapists and other members, showing aggressive, conflicting and incoherent behaviour; the work group, differently from other basic assumption groups, is based on good cooperation between participants on a topic, theme or problem to be solved, without the interference of strong emotions or destructive conflicts, showing the participants' ability to cooperate and control their emotions.

 3) The quantitative analysis seems very opaque to me; from the descriptions provided by the authors, it is not possible to gather any information about the questionnaire, how many items it entailed, what constructs were assessed; about the statistical analyses, not enough details are provided to reproduce the procedures. This also makes it impossible for me to understand Figure 1. Further, it seems like a larger number of regression models was conducted, but only the significant results are reported. To me, this seems to lack adherence to general principals of good scientific practice in quantitative research methods. By conducting multiple analyses, the alpha error is increased and the p-value should be corrected by this; if the analysis is intended as an exploratory analysis, this may not be needed, but analyses should be marked as such and all results, irrespective of significance, should be reported.

Response to Reviewer 2: Thank you for your meaningful remark. We added more information about MBT-G-AQS in the Method section in order to better understand Figure 1.

In accordance with your suggestion, we chose only one model of multivariate regression between overall occurrence score of MBT-G-AQS and the other selected variables, as specified in the Method section. Table 6 was modified.

4) How were the narrative themes investigated, who decided for these and by which criteria?

    Response to Reviewer 2: Thank you for your thoughtful questions. We added the following description in the Method section regarding our thematic analysis as you suggested:

 “2.            Thematic analysis related to the main narrative nuclei, using an inductive approach in 5 phases: 1) becoming familiar with the topic, 2) creation of initial codes, 3) iden-tification of the main themes, 4) qualitative review of the main themes, 5) definition and naming of final themes. The main narrative cores were identified by thematic analysis, i.e., a systematic method for identifying, organizing and investigating themes within a data collection by using an inductive approach [27]. We conducted the thematic analysis on the collection of data recorded by the therapists at the end of each group session.

     Overall, the thematic analysis was conducted through 5 stages by the therapists

     who participated in group session:

Step 1. becoming familiar with the topic: we proceeded by analytically re-reading  expressed themes, becoming familiar with them in order to identify the relevant ones;

Step 2. creation of initial codes: codes can be defined as a kind of label or concise summary of the expressed themes created by interpreting both semantic and latent contents. The initial codes created were the following: inside/outside relating to the internal or external environment of the department; mistrust in the healthcare sys-tem; stigma of mental illness; mental suffering; perception of the passage of time; disease awareness; doctor-patient relationship; meaningful interpersonal relation-ships; emotion regulation; search for daily recreational activities; paranoid ideas; hospitalization experience; somatizations and bodily concerns;

Step 3. Identifying the main themes: a theme is defined as a significant central concept or idea that recurs within multiple topics. Generating main themes was an active process of re-viewing initial codes, identifying areas of similarity or overlap between them, generating subthemes, and bringing together codes that appear to have a common characteristic so that they can describe a consistent pattern within all codes;

Step 4. Review of the main themes: themes were reviewed assessing their relationship to all other themes and their ability to synthesize the most relevant and important elements in relation to the research question;

Step 5. Definition and naming of main narrative themes: any emerging narrative nuclei were named based on the following 5 themes: Interpersonal relationships; Healing process; Introspective experiences; Paranoid ideas; Daily life activities. In some sessions, multiple main narrative themes emerged. Therefore, we grouped multiple main narrative themes together into the three following ones: A) Treatment programs + Interpersonal relationship; B) Treatment programs + Introspective experience; C) Interpersonal relationship + Introspective experience. Finally, we collected a total of 8 main narrative themes .

2.4. Psychometric analysis for mentalization

We measured the level of mentalization in the group by completing the Mentalization-Based Therapy-Group-Adherence and Quality Rating Scale (MBT-G-AQS), which  evaluates 19 items in terms of occurrence and quality [27]: 1) Managing group boundaries, 2) Regulating group phasing, 3) Initiating and fulfilling turn taking, 4) Engaging group members in mentalizing external events, 5) Identifying and mentalizing events in the group, 6) Caring for the group and each member, 7) Managing authority, 8) Stimulating discussions about group norms, 9) Cooperation between co-therapists, 10) Engagement, interest and warmth, 11) Exploration, curiosity and not-knowing stance, 12) Challenging unwarranted beliefs, 13) Regulation of emotional arousal, 14) Acknowledging of good mentalization, 15) Handling pretend mood, 16) Handling psychic equivalence, 17) Focus on emotions, 18) Stop and rewind, 19) Focus on the relationship between therapists and patients. For items n. 6, 7, 10, 13 and 15 only quality assessment is required.

Occurrence (score from 0 to 30) indicates how often therapists recall the mentalization process to patients in the group session; quality (score from 1=very poor to 7=excellent according to a 1-7 Likert scale; 0=not applicable) indicates the level of skill and expertise of the therapist in handling the item content during the group session. The overall rating of occurrence and quality should indicate the global functioning of the group as ground for mentalizing [27].”

5)   In the discussion, the authors communicate a sense of positive surprise; if I read this right, given the high fluctuation and strong variance in diagnoses in the psychotherapy group within the acute psychiatric ward, it might be expected that groups would show little coherence and distrust towards the practitioner might be predominant, but on the contrary, the results demonstrate a fruitful working environment. This is a finding inspiring trust in therapy groups in this setting and I appreciate it. However, mentioning these expectations only in the discussion seems to be a waste, I expect that the manuscript could benefit if the authors communicated their anticipations already in the Introduction. Typically, one would expect a hypotheses-section containing such expectations.

Response to Reviewer 2: Thank you for your thoughtful suggestion. We added the following hypotheses-section as you suggested after Objectives of the study.

Expected results

      We took into account the main features of subjects hospitalized in an acute ward: acute psychopathological conditions, rapid turnover of participants due to brief hospitalizations, the heterogeneity of inpatient psychopathology and diagnosis, the variability of psychopharmacological treatments, as well as the adherence to treatment. Due to these characteristics, we expected narrative themes dominated by paranoid contents, suspiciousness and oddity, a prevalence of oppositional and disorganized groups ac-cording to Bion’s classification and approximately 50% of inpatients participating in the group therapy with reduced mentalization process capacity.”

6.) The authors report that a peer support person helped the group maintain a positive work environment. While I support this notion and could imagine very well that this is a helpful factor, no comparison group without the peer is available. Thus there is no way of telling how much of the group’s characteristics were fostered by the presence of the peer. If no comparison group is available, this cannot be helped, but should be discussed.

Response to Reviewer 2: Thank you for your thoughtful suggestion. We  totally agree with you. Therefore, we changed the Conclusion in the abstract and modified the following sentence in the Conclusions in the main manuscript.

“We presume that the involvement of a PSP might have contributed to the regular group activity also in a setting shaped by urgencies, acting as an intermediary with the outside world and, at the same time, maintaining the so-called memory of group sessions.”

We also reported your observation among the limits of the study:

     “Finally, one of the limitations of the study is the lack of a control group without the peer support which could have highlighted the role of this figure in preserving the continuity of the group setting, fostering the stability of group psychotherapy.”

In summary, it appears to me that this manuscript might benefit greatly from trimming it and removing concepts and analysis parts that are not essential in favor of the more important parts, which could then be described much more appropriately.

Comments on the Quality of English Language

To my mind, the use of scientific English was adequate; minor spell and punctuation checks will be sufficient. 

I have noticed lacking commas in enumerations throughout the manuscript ("a, b, and c"; or "a, b, or c").

Response to Reviewer 2: We checked the language and the punctuation with the support of a native English speaker.

Thank you for having taken so much time to carefully review our manuscript. I hope that the changes we have made are sufficient to improve the manuscript and make it acceptable for publication.

Best regards

Rosaria Di Lorenzo

Round 2

Reviewer 2 Report

I think that the additional explanations and clarifications provided by the authors help a great deal in making the article more understandable. I have a few more remarks: I suggest 1) adding information as to the source of the data to the Abstract, but also to later parts of the manuscript. In the Abstract, the authors write that they “collected 40 weekly group sessions” – from the outside, it is not clear what is meant. Data were collected about the group sessions, but the reader does not know who provided the data: were the healthcare professionals interviewed or asked for their judgment? Were the patients asked? This applies also to the questionnaire data, it does not become clear from the article who filled in these questionnaires. 2) I suggest rephrasing the last sentence of the Abstract, it seems unclear what is meant by a unicum – I suggest rephrasing the conclusion in clearer terms. 3) In the section where the authors describe the group types characterized by Bion (p. 4-5), the punctuation and sentence structure could be changed to improve readability; e.g., some terms describing groups are written in quotation marks, others are not. Furthermore, many sentences and sentence parts are separated by semicolons. I suggest shortening the sentences, separating them by periods, and using clearer words (the xy group is defined as...).

To my mind, my previous comment was not fully addressed, still there seem to be commas lacking; possibly, MDPI will offer editing services? 

Author Response

Response to Reviewer 2

I think that the additional explanations and clarifications provided by the authors help a great deal in making the article more understandable.

Thank you for your positive comment.

 I have a few more remarks: I suggest

1) adding information as to the source of the data to the Abstract, but also to later parts of the manuscript. In the Abstract, the authors write that they “collected 40 weekly group sessions” – from the outside, it is not clear what is meant. Data were collected about the group sessions, but the reader does not know who provided the data: were the healthcare professionals interviewed or asked for their judgment? Were the patients asked? This applies also to the questionnaire data, it does not become clear from the article who filled in these questionnaires.

Response top Reviewer 2.

Thank you for your suggestion. We modified the paragraph of Methods in the Abstract as follows:

“2) Methods: In 40 weekly group sessions held in an acute psychiatric ward during one year, we retrospectively evaluated the inpatients’ participation and demographic and clinical variables of individuals hospitalized in the ward,  the group type according to Bion’s assumptions, the main narrative themes expressed and the mentalization processes by using Mentalization-Based Therapy-Group Adherence and Quality Scale (MBT-G-AQS);”

2) I suggest rephrasing the last sentence of the Abstract, it seems unclear what is meant by a unicum – I suggest rephrasing the conclusion in clearer terms.

Response top Reviewer 2.

Thank you for your suggestion. We rephrasing the last sentence of the Abstract as follows:

“4) Conclusion: our study suggests that group shows consistent defense mechanisms, relation-ships, mentalization and narrative themes, which can maintain a therapeutic function also in an acute ward.”

3) In the section where the authors describe the group types characterized by Bion (p. 4-5), the punctuation and sentence structure could be changed to improve readability; e.g., some terms describing groups are written in quotation marks, others are not. Furthermore, many sentences and sentence parts are separated by semicolons. I suggest shortening the sentences, separating them by periods, and using clearer words (the xy group is defined as...).

Response top Reviewer 2.

Thank you for your suggestion. We changed the punctuation and sentence structures as you suggested:

  1. Analysis of the functioning mode of the therapeutic group according to Bion's classification [6]:
  • “the fight-flight group” or oppositional group, occurs when group participants ally against the therapists identified as the cause of their conditions, consider hospitalization as a detention, show mistrust towards the therapeutic processes and/or criticize the rules of ward;
  • ”the dependency group” or psychoeducational group occurs when the group participants ask for health information (clarification on psychiatric pathologies, length of hospitalization, drugs and their side effects, criteria for compulsory medical treatment, etc.) and need to be continuously reassured and supported by therapists, showing passive and immature behaviour;
  • “the pairing group” or psychological confrontation group is characterized by hoping and magical waiting for rescue through two parties uniting to create the perfect solution for participants’ current conditions, which they are not able to ac-tively modify; the disorganized group in which participants, often suffering from an acute psychiatric condition, are unable to find adequate verbal communication and an effective relationship with therapists and other members, showing aggressive, conflicting and incoherent behaviour;
  • “the work group”, differently from other basic assumption groups, is based on good cooperation between participants on a topic, theme or problem to be solved, without the interference of strong emotions or destructive conflicts, showing the participants' ability to cooperate and control their emotions;

Comments on the Quality of English Language

To my mind, my previous comment was not fully addressed, still there seem to be commas lacking; possibly, MDPI will offer editing services?

The punctuation was revised and some commas were added in the manuscript.

Best regards

Rosaria Di Lorenzo
